# Stereo Camera Setup for 360° Digital Image Correlation to Reveal Smart Structures of *Hakea* Fruits

**DOI:** 10.3390/biomimetics9030191

**Published:** 2024-03-21

**Authors:** Matthias Fischer, Max D. Mylo, Leon S. Lorenz, Lars Böckenholt, Heike Beismann

**Affiliations:** 1Westfälische Hochschule, Münsterstraße 265, 46397 Bocholt, Germany; matthias.fischer@w-hs.de (M.F.); lars.boeckenholt@studmail.w-hs.de (L.B.); 2Cluster of Excellence livMatS @ FIT—Freiburg Center for Interactive Materials and Bioinspired Technologies, University of Freiburg, Georges-Köhler-Allee 105, 79110 Freiburg im Breisgau, Germany; max.mylo@livmats.uni-freiburg.de; 3Department of Microsystems Engineering—IMTEK, University of Freiburg, Georges-Köhler-Allee 078, 79110 Freiburg im Breisgau, Germany

**Keywords:** biomimetics, functional morphology, plant biomechanics, plant motion, strain analysis, structure–function relationship, 3D digital image correlation (3D-DIC), *Hakea sericea*, *Hakea salicifolia*

## Abstract

About forty years after its first application, digital image correlation (DIC) has become an established method for measuring surface displacements and deformations of objects under stress. To date, DIC has been used in a variety of in vitro and in vivo studies to biomechanically characterise biological samples in order to reveal biomimetic principles. However, when surfaces of samples strongly deform or twist, they cannot be thoroughly traced. To overcome this challenge, different DIC setups have been developed to provide additional sensor perspectives and, thus, capture larger parts of an object’s surface. Herein, we discuss current solutions for this multi-perspective DIC, and we present our own approach to a 360° DIC system based on a single stereo-camera setup. Using this setup, we are able to characterise the desiccation-driven opening mechanism of two woody *Hakea* fruits over their entire surfaces. Both the breaking mechanism and the actuation of the two valves in predominantly dead plant material are models for smart materials. Based on these results, an evaluation of the setup for 360° DIC regarding its use in deducing biomimetic principles is given. Furthermore, we propose a way to improve and apply the method for future measurements.

## 1. Introduction

Fruits of the native Australian species *Hakea sericea* and *Hakea salicifolia* of the Proteaceae family possess a special opening mechanism to release their seeds. When the sap connection to the mother plant is severed (e.g., by a bush fire), the fruits dry out and shrink. During this process, their valves deform and separate along predetermined breaking tissues at the ventral suture and dorsal side [1]. The opening along both sides is unique within the Proteaceae and is only found in the genus *Hakea* [2]. The ventral suture is characterised by a preformed rupture line, with tissue that lies very loosely between the two sides of the carpel. Opening across this suture should not offer any resistance and should be possible without force. On the dorsal side, a highly lignified tissue comprised of smaller and differently oriented cells reveals the second predetermined rupture line of the fruit. A certain amount of force is required to propagate a crack in this area [3]. These fruits can be considered smart biological structures because they show an autonomous reaction to an external stimulus. In this way, the plants ensure that fruits open only when environmental conditions allow successful seed dispersal, which may well be after several years. In their natural habitat, these conditions are, for example, met after a bush fire event, when germination conditions are optimal due to the absence of competition and fertilised soil from the ash.

Besides being of great importance in natural systems, predetermined breaking points and self-actuated movements play an important role in many technical components, e.g., bursting discs, airbags, electrolytic capacitors, (emergency) valves or self-locking building materials. A possible biomimetic transfer to technical solutions is, therefore, the motivation to understand the opening mechanism more precisely.

Both the fracture mechanism and the actuated movement of the two valves in predominantly dead plant material are, therefore, models for smart adaptive materials and structures. According to the biomimetic development process [4,5] the transfer to a technical solution can only be successful if the biological models are understood as fully as possible.

Due to the size (a few centimetres in diameter) and complex shape of the fruits, digital image correlation (DIC) is an ideal method to determine strain and shrinkage characteristics of the fruits’ surfaces [6]. However, using a single perspective stereo camera setup would only reveal parts of the complex geometry and inhomogeneous, anisotropic shrinkage of these fruits during opening (Figure 1). Therefore, this paper will first explain how DIC has been used to analyse displacements and deformations on the surfaces of moving objects. It then discusses the shortcomings for complex geometries and how these can be overcome for a specific problem using the method presented.

Digital image correlation (DIC) is a non-contact optical technique for measuring full-field displacement and the strain of surfaces. DIC evolved from classical photogrammetry with the advent of digital cameras in the early 1980s and has been continuously improved ever since [7,8,9,10,11]. Its first applications were in materials science, but over the years, DIC has been utilized more and more in other research fields, such as aerospace, including large-scale composite structures [6], civil engineering, such as bridge monitoring [12,13], human and animal biomechanics to analyse a wide variety of organs and tissues [14,15] and wood research, from entire trees to processed composites [16]. In recent years, DIC has also been used in plant analysis, namely to analyse plant tissue strains under tensile loading (e.g., mistletoe–host interface [17] or the branch–branch connections of cacti [18]) to analyse compression (e.g., citrus peels [19]), and to characterise plant movements, such as the snap-buckling closure [20] and reopening [21] of the Venus flytrap or the desiccation-driven motion of pine cone scales [22,23]. This has led to a better understanding of the functional principles of the plant material systems, which is a prerequisite for successful transfer to bioinspired materials.

In general, there are two types of DIC systems: single camera (2D-DIC) and stereo camera (3D-DIC). In a 2D-DIC setup, a single camera is positioned perpendicular to the surface being analysed, said surface must be flat and the deformation must take place in-plane [24]. To analyse the displacement or strain of uneven surfaces, or out-of-plane motions, a 3D-DIC setup is necessary. This requires two cameras which are oriented toward the specimen at a defined stereo angle [25]. Not only does standard 3D-DIC carry additional costs for a second camera, but it also requires synchronisation of the two cameras and calibration of the measurement volume.

The core principle of DIC, whether for 2D-DIC or 3D-DIC analysis, is the partitioning of images of a sample’s surface into small areas called subsets or facets [24,26]. Each subset of pixels has a distinct grey value distribution. These grey value distributions of a reference image (usually of the unstressed/undeformed sample) are then compared with the grey value distributions of images of the deformed/moved sample (usually several hundred images that track the process of deformation/movement). This is completed in a defined search field centred on the original position of the subset. Correlation algorithms are used to determine the best match of the subset grey value pattern, and the subset is placed at the location in the deformation image with the highest resulting correlation score [24,26]. By mapping the resulting displacement vectors of all subsets, a displacement field of the surface of interest is generated. A surface strain pattern can then be calculated by comparing adjacent displacement vectors. The method itself has no temporal resolution limit, however, the cameras used to capture the images do. Therefore, depending on the camera type and speed, DIC can analyse very fast motions and deformations [20], as well as relatively slow motions over days using time-lapse recordings [22]. In addition to obtaining sharp, high-resolution stereo images, the basis for accurate displacement and strain analysis with a high spatial resolution is a high level of surface contrast [27,28]. In most applications, this is achieved by applying a random pattern of black speckles to a white background (e.g., using a spray can or airbrush). DIC can also analyse various sample sizes ranging from a few millimetres to several meters.

One of the main limitations of DIC is that, even with a stereo camera setup, it has a restricted range of analysis—3D-DIC requires that the object’s surface is visible to both cameras (Figure 1).

If the region of interest (ROI) on a sample is highly curved, it may not be included in the field of view (FOV) of a single stereo camera system. Only in recent years, solutions have been developed to extend the FOV of the system to 360° of the sample surface (to our knowledge, the group of Degenhardt were the first who applied 360° DIC in 2007, although the term DIC is not mentioned in the related papers [29,30]). While several different solutions have been proposed for multi-view/panoramic/360-degree or omnidirectional DIC, these solutions all share that they use, either physically or virtually, more cameras than the standard stereo camera (3D)-DIC setups.

One method for incorporating more than two cameras into the DIC system exploits multiple-image triangulation and, thus, requires a high degree of overlap between all FOVs of the cameras involved. According to this principle, Harvent et al. and Orteu et al. built a four-camera system that, in contrast to an earlier two-camera attempt, allowed for 3D reconstructions and subsequent measurements of displacements of the entire upper surface of a sheet metal while it underwent a forming process [31,32].

Larger FOVs are more easily obtained by adding pairs of (binocular) stereo cameras. Here, the views only need either a small overlap to provide a continuous measurement surface, or no overlap if the FOVs are distributed [33,34]. In contrast to multiple-image triangulation, these camera pairs are calibrated separately using either direct linear transformation (DLT) or bundle adjustment (BA) algorithms [26,35]. The DLT technique places a three-dimensional (non-planar) calibration target, with a pattern of calibration points, in the measurement space. The precise knowledge of the spatial positions of these points in a global coordinate system is necessary. Every camera takes a single picture for each position, which allows for the determination of intrinsic (e.g., focal length) and extrinsic (e.g., the camera’s position and orientation) parameters of each camera pair [33]. Since this method is based on a simple pinhole camera model, the distortion parameters of the cameras have to be determined by other methods before applying DLT calibration [33]. As in the above-mentioned multiple-image triangulation, surface data calculated from each DLT-calibrated camera pair is expressed in global coordinates [33]. BA requires an independent calibration of each stereo camera pair by taking images of a flat calibration target, that carries a dot grid pattern, in different orientations. The method yields all intrinsic and extrinsic parameters of a single stereo camera system, including distortion parameters [26]. However, spatial data of 3D-reconstructions are only expressed in local coordinates of the respective stereo camera system. Transforming the data into global coordinates is called stitching and may be achieved by calculating transformation matrices from at least three reference points in overlapping areas. In these cases, a master stereo camera system is selected. Here, the local and global coordinate systems are the same and all other stereo camera systems are aligned to it [33]. Solav et al. used BA to calibrate all stereo camera systems to obtain an estimation for their distortion parameters. They then used the distortion-free images for a DLT calibration [33,36].

Stereo camera pairs can be arranged in a variety of ways based on the requirements of the test object. For example, cameras can be arranged as a wall when analysing wind turbine blades or concrete beams [37,38]. To analyse large structures, such as an arch-shaped hall made of self-supporting metal plates, they can be broadly distributed [34]. Face-to-face cameras function well for standard tensile tests and a surrounding 360° setup works well for biological in vitro or in vivo analyses [36,39,40].

Capturing panoramic (360-degree) views of a sample’s outer surface requires minimally five cameras, but the exact number can be adjusted depending on the shape of the sample surface [41]. The literature describes a few setups that consist completely of real cameras for panoramic DIC (pDIC = 360° DIC), as shown in Solav et al. and Sun et al. [36,42]. Cost reduction as well as insufficient space around a sample for a multi-camera setup are reasons for the development of pDIC setups with virtual cameras [41].

A simple implementation of virtual cameras for 360° DIC measurements consists of a real camera or stereo camera pair that moves stepwise in a circle around the fixed sample, taking images at each step [43]. In an equivalent approach, the position of the camera (pair) is fixed and the sample rotates stepwise [39,44]. Approaches that arrange (virtual) cameras in a circle around the sample usually combine all their contiguous positions as stereo camera pairs, so that each (virtual) camera is part of two stereo sensors [33,41]. Badel et al. took images with a video camera while the sample rotated. This achieved a high density of 200 evenly distributed virtual cameras around the sample; remarkably, only 8 of these were combined as stereo sensors [44]. Since both equivalent virtual camera techniques require several sub-steps to collect images for a single panoramic measurement, only samples that do not deform or move markedly during the time required for these image collections are suitable [45].

Other virtual camera setups for 360° DIC use mirrors to capture several different views of a sample in a single image [46]. The research group of Pan developed and improved a setup consisting of a stereo camera pair and two mirrors. With this setup, it is possible to measure lateral surfaces of cylinders or both sides of a flat bar for tensile testing [47,48,49,50,51,52,53]. By further splitting the view with additional mirrors, a single camera is sufficient for 360° DIC measurements [54]. To collect all data into a global coordinate system, a reflection transformation must be performed on the reconstructed mirror views. The most convenient way to achieve this transformation is a calibration using speckle-patterns on both mirrors [46]. Since all views share the same camera sensor, the highest possible resolutions are reduced compared to a mirror-free setup where only one view (+calibration target) occupies the entire sensor.

Genovese et al. developed a pDIC method to measure strains of arteries and other organs in vitro in physiological salt solution [55,56,57,58]. Their method combines the use of mirrors with the technique of camera movement. More precisely, a high-precision conical mirror and a single camera were used to map the entire lateral surface of a sample into a single warped image. For stereo information, the camera view was slightly angled towards the bisecting line of the conical mirror (e.g., 1°). Several images were then taken from different positions while the camera view rotated at the same angle around the bisecting line. This approach requires that the images are corrected using a calibration target and refractions correction. Additionally, the size of the conical mirror limits the sample size to a few centimetres in diameter and height. Genovese et al. also proposed 360°-DIC setups using a stereo camera pair and a conical mirror [59], as well as a setup employing refraction to create virtual cameras [41].

So far, 360° DIC has been used to measure displacements/strains of biological samples of animals or humans in vitro (e.g., the mentioned publications by Genovese et al.) and in vivo [33,36,51], technical materials [49,50,60] and, as the only plant example, a compression test of a timber column [40]. Non-circular multi-view DIC methods are predominantly used for measuring larger scale engineering structures [34,37,38,61,62,63]. To the best of our knowledge, multi-view DIC has neither been tested to measure self-actuated movements of plants, nor with the aim to transfer natural structures to technical solutions.

The aim of this work is to establish and validate a novel approach for 360° DIC. We achieve this by using a stereo camera pair of a commercial 3D-DIC system that generates virtual camera pairs by a continuous stepwise sample rotation. We evaluate our 360°-DIC method based on the completeness of the obtained surface data during all recorded stages of the fruit opening. This method was chosen because complete surface data are necessary for fully determining the displacements and strains of the natural material’s surfaces. With this data, we can subsequently deduce characteristics of smart adaptive structures. As proof of concept of this setup, we track the displacements of the surfaces of *H. sericea* and *H. salicifolia* fruits during their desiccation-driven opening. Furthermore, we propose a way to improve and apply the method for future measurements.

## 2. Materials and Methods

The ARAMIS AdjusTable 12M stereo camera system (Carl Zeiss GOM Metrology GmbH, Leipheim, Germany) are used for all measurements. The two integrated cameras have a resolution of 12 megapixels (4096 × 3000 pixels with a pixel size of 3.45 µm); they are each equipped with a 50 mm Xenon Jade 2.8/50 lens (Schneider-Kreuznach, Bad Kreuznach, Germany) and a linear polarisation filter. Samples are illuminated with linear polarised blue light. Images are recorded as 8-bit greyscale images. A standard stereo camera angle of 25° is used as recommended by the manufacturer. Measurements and post-processing are performed using the 2021 version of the GOM Correlate Pro software from the same manufacturer on an integrated analysis computer. The system is equipped with its own controller, allowing image acquisition via external triggers.

In order to perform 3D-DIC imaging, the camera system must be calibrated. For this purpose, a calibration object (CP40/MV60/21387 from Carl Zeiss GOM Metrology GmbH) is used, and images are taken in nine different positions and orientations (in accordance with the GOM Correlate Pro software instructions). From these stereo images, the intrinsic and extrinsic camera parameters are determined by the software. Care is taken to ensure that the position and orientation of the cameras are not disturbed after the calibration process.

All measurements are conducted with fruits of *H. sericea* (see Figure 2) and *H. salicifolia* that are at least 2-year-old fruits. These were collected on 1 September 2022 near Coimbra, Portugal (N 40°10.64137 W 8°27.6702). To avoid unintentional opening during storage, the samples were kept moist and in the fridge at 5 °C.

To prepare the fruits for measurement, the epidermis and the underlying bark are removed down to the wood (Figure 2). According to preliminary experiments, neither the epidermis nor the bark contributes to the opening mechanism, but rather prolongs the desiccation time and leads to cracks and hollow bulges during the drying process. By removing the epidermis and bark, it is possible to generate a surface to which the speckle pattern adheres well and to achieve a shortened opening process of one to two days.

To create a speckle pattern with a high-surface contrast, the peeled fruits are first primed with white acrylic paint (Müller Hobby Acryl Mattfarbe, Müller, Bundesland, Germany) before fine random speckles are applied using black spray paint (RENOVO Sprühlack Rallye Matt, Hagebau Gruppe, Lower Saxony, Germany) (Figure 2). This artificial stochastic pattern is a basic requirement for reliable surface detection with any DIC system. The prepared fruits are stuck with their peduncle to a rotating sample holder by means of a putty-like adhesive (Blu Tack, Bostik GmbH, North Rhine-Westphalia, Germany).

## 3. Results

### 3.1. Experimental Setup for 360° DIC with One Stereo Camera System

The experimental setup for 360°-DIC measurements utilises the ARAMIS Adjustable 12M stereo camera system including an analysis computer and a control unit. The setup also includes a rotatable table where the sample is fixed to and control electronics with a built-in Arduino UNO Rev3 (www.arduino.cc, accessed on 20 March 2024). The rotating table consists of a laser-cut plywood disc moved by a stepper motor. In addition to the sample, a 3D-printed frame with randomly distributed reference point markers (0.8 mm diameter) is mounted on the rotatable table (Figure 3). The heat-resistant material acrylonitrile styrene acrylate (ASA) is chosen for the 3D-printing in order to create a dimensionally stable reference point frame. To initially obtain a 3D representation of the reference point locations in space (reference point cloud), images are taken with the stereo camera pair looking from above by tilting the table, to capture all reference points simultaneously with both cameras. Reference points are automatically identified by the GOM Correlate software, and their positions are calculated by triangulation based on the calibration information. In the subsequent measurements, each image is precisely positioned in the global coordinate system, according to the reference point cloud. For this reason, at least three common reference point markers must be visible in each stereo image pair while additional visible point markers increase accuracy. We used a total of 48 reference point markers on the frame, with at least 10 reference point markers visible in each image pair.

The control electronics are wired to the stepper motor and the “Trigger In 0” input of the ARAMIS controller. This connection transmits signals for generating images to the camera system. The built-in Arduino UNO Rev3 controls the rotation of the table, including the number of individual steps for a complete 360° rotation, the pause at each position and the pause after a full rotation. The stepper motor has a resolution of 200 steps per 360°. Minor inaccuracies of the rotation angle inside a single step are compensated by the alignment to the global coordinate system with the reference point markers. Consequently, the alignment of sample and stereo camera pair does not need to be exactly the same at each time a virtual stereo sensor captures an image, and the spatial accuracy depends solely on the accuracy of the triangulation of the reference points. Large deviations in the alignment, on the other hand, would result in a loss of surface detection in the DIC process.

### 3.2. 360°-DIC Measurement

Preliminary tests revealed that at least eight positions are needed to achieve sufficient overlap between the adjacent views for an almost complete coverage of the surface. Therefore, a full 360° rotation is divided into eight equal steps of 45°. At each of these eight recording positions, the acquisition of an image is triggered by the Arduino via the “Trigger In 0” input line of the ARAMIS controller. Thereby, eight virtual stereo sensors are created which correspond to the eight angular positions that are captured (Figure 4). After completing a rotation in 13.6 s, the system pauses for ten minutes before starting a new rotation. The resulting measurement frequency was found in preliminary tests to be sufficient to capture the opening mechanism of *Hakea* fruits, but the intermittent pause can be adjusted according to the velocity of the monitored movement of an object. All of the images generated during a rotation are assigned to the same time step. The same angular positions are used for each rotation and all of the images generated at the same angular position are assigned to the corresponding virtual sensor.

In order to create a superimposed surface model based on the different image positions within a time step, all local surface models have to be transferred into a common coordinate system. This is achieved by capturing the reference points around the sample in recording and detecting them with the GOM Correlate Pro software to reconstruct their positions based on the obtained reference point cloud. This allowed for precise positioning and alignment of the captured images to the common coordinate system (Figure 5).

The data from every virtual sensor are then processed individually for 3D reconstruction and 3D-DIC to produce a corresponding part of the surface model, called a surface component (see following section and Figure 6). The overlay of all parts results in a complete surface model (Figure 7).

### 3.3. 360° DIC Results

After successfully orientating the images in the global coordinate system and assigning them to the corresponding virtual sensors, the surface model is created as a basis for further displacement and strain analyses. To achieve this, a surface component is generated for each virtual sensor and then merged with its neighbours. Figure 6 shows this, using the example of two surface components obtained from different viewing perspectives. The first and second surface components are created from the last set of stereo images of virtual sensor 1 and virtual sensor 2, respectively. Together, these two surface components produce the surface model shown in the right image panel.

By generating the surface components from data of all eight virtual sensors, an almost-complete surface model can be created for the monitored part of the opening process of both species (Figure 7). Overlapping areas of surface meshes built from data of adjacent virtual sensors are almost identical. A systematic assessment of the spatial accuracy has not yet been performed. However, cross-sections of the overlapping surface meshes indicate that the position of a point on the fruit’s surface has a maximum deviation of about 0.1 mm between the different meshes. Therefore, all data are included in the calculation of displacements. Only surface components that are furthest outwards are visible in the surface models shown in Figure 7.

However, some surface areas at the top (distal side) and the bottom (proximal side) of the fruits, as shown in Figure 8, are not visible to at least one camera in either orientation. This is caused by undercuts and strong changes in curvature, resulting in a lack of surface detection in these areas.

During the measurement period, areas of the fruit that are initially visible also disappear from the cameras’ field of view from time to time. This slightly reduces the completeness of the surface model. Overall, however, the surface model is largely complete throughout the monitored opening process and is, therefore, suitable for further analyses. All surface components displayed in the software can be analysed with regard to their displacement (and strain) behaviour during fruit opening. The first time step is used as the reference stage. The desiccation-driven displacements are colour-coded in the surface model and displayed on a scale. As shown in Figure 9 for *H. sericea*, the measured displacements increase throughout the measurement period, with the highest displacements occurring at the tips of the two fruit halves. At the end of the measurement, the displacement in these areas is up to 12.9 mm. Towards the base of the fruit, the measured displacements decrease uniformly. At the fruit base, hardly any displacements occur. Both fruit valves experience comparable displacements. Further displacement and strain calculations can now be carried out for the surface models of both species *H. sericea* and *H. salicifolia*.

## 4. Discussion

Biological structures that deform in space and actuate a movement upon an external trigger [23,64,65,66] are role models for smart adaptive materials. This is particularly true for dead plant material that responds to external stimuli since it does not depend on the complex metabolism of living cells. *Hakea* [67] or *Banksia* [64] fruits are examples for the latter principle. They form a seed bank in the canopy, and the fruits only open when an external stimulus indicates favourable conditions for seed germination.

The functional structures of these fruits are examples for actuators with increased longevity [68]. Longevity is a desired property of smart materials that need to react to a trigger reliably, even after years in a dormant state. Many biological role models for actuated moving structures are functional long-term. The most impressive example is the moisture-controlled opening and closing of pine cones to release their seeds in dry weather, even after thousands of years [69].

Because the presented method is a non-contact technique that provides full-field displacement and strain measurements with an almost-complete surface coverage, the proposed 360° DIC is particularly suitable to determine the strains and movements of self-actuated sample deformations. Correlations between the actuation (in our case, humidity) and sample movements can be analysed. In a following step, deviated kinematic data deliver the frame for the adaption of material models. One way to complete the transfer from the natural principle to a technical solution is the subsequent 4D-printing of prototypes. In this context, more and more work is currently being completed with the aim of developing self-shaping components or materials [70,71]. 

It should be noted that the artificial application of a speckle pattern could clog the pores on the surface of the fruit, reducing evaporation and, thus, slowing the opening movement. In our example of *Hakea* fruits, the opening time is not comparable to natural conditions anyway, due to the removal of the epidermis and bark, so quantification should be omitted here. However, it can be assumed that the kinematics of the opening is not influenced by blocked pores, despite the slowed dehydration, as this is structurally predetermined. This was shown, for example, by Correa et al. [22] for the drying movement of sprayed pine scales, where a deceleration but no change in kinematics was observed. With the presented DIC method, almost complete surface models of objects (here fruits of *H. sericea* and *H. salicifolia*) can be generated with a single stereo camera system using a rotatable table to capture a 360°-view of the sample. In addition to the rotating table, a key part of the measuring method is the calibration with a reference point cloud via a reference marker frame. This calibration allows reconstructed spatial data of all virtual camera pairs (virtual stereo sensors) to be expressed in coordinates of a common coordinate system. By combining the spatial data, a superimposed surface model can be generated for each time step of the measurement. An object that exhibits a particular motion, e.g., the opening of a fruit, can now be analysed over the entire motion sequence at different time steps.

The method still requires a systematic error assessment. An observed maximum error of about 0.1 mm between the coordinates of the same surface point in the meshes of different sensors, often referred to as merging error or stitching error, indicates an accuracy of the same order as those of other 360°-DIC methods [33,62].

The temporal resolution of the presented 360°-DIC system based on a moving sample is determined by the time required for a complete image acquisition of all camera positions of one rotation. With our setup, it takes 13.6 s to record a full rotation with 8 steps. During this period, the sample is supposed to remain approximately in the same state, making this method especially feasible for relatively slow movements/deformations, such as those frequently found in biological desiccation-driven movements.

However, the proposed method is currently limited by the fact that some parts of the fruit surfaces remain undetected due to the circular orientation of all virtual camera pairs in one single plane (Figure 8). Due to strong movements or deformations of the sample, other parts of the surface may move out of the range of the camera sensors in later time steps, making them undetectable for the stereo camera pairs. Generally, the completeness of the surface model can be improved by the addition of further virtual camera pairs that do not lie within the circular plane. This could be realised by a mechanism that tilts the rotating table by a certain angle, so that a circle of images with a horizontally oriented sample is followed by a corresponding circle with a tilted table and sample.

Additionally, the method is currently not readily applicable to 360°-DIC recordings during artificial force application, such as in tensile or compression experiments, because the testing machine would need to be rotated and parts of its rods may block the view on the sample in some orientations. However, if applicable, the method provides a less complex alternative with lower equipment and space requirements when compared to other published multi-camera systems.

In summary, with the presented method, it is possible to measure the displacements and subsequently determine the compressions or strains that occur at the surface, as shown for two species of *Hakea*. This is the first time a 360°-DIC method has been used to monitor self-actuated movements of a plant. The setup of the presented 360°-DIC method consists of known elements in a new combination and was specifically designed to capture relatively slow movements. The data can now be used to adapt or validate a plant tissue-based finite element model, as shown for the Venus flytrap [20] using surface models or for bamboo using X-ray tomography as image sources [72], to predict involved forces and subsequently lead to simplified material models that mimic the deformation and/or the fracture via a predetermined line. This will result in a method to subsequently generate data for a deeper insight into the opening mechanism of this biological model. From this source, a biomimetic development process can be used to translate the findings into technical solutions.

## Figures and Tables

**Figure 1 biomimetics-09-00191-f001:**
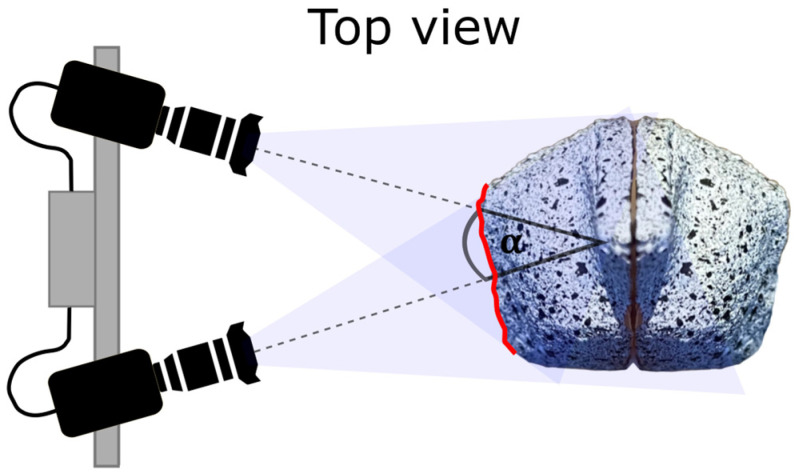
Schematic representation of the visual limitation of a conventional stereo camera system (with stereo angle α) for samples with strong curvature, taking a *Hakea* fruit as an example. The surface of the fruit covered by both cameras is marked in red.

**Figure 2 biomimetics-09-00191-f002:**
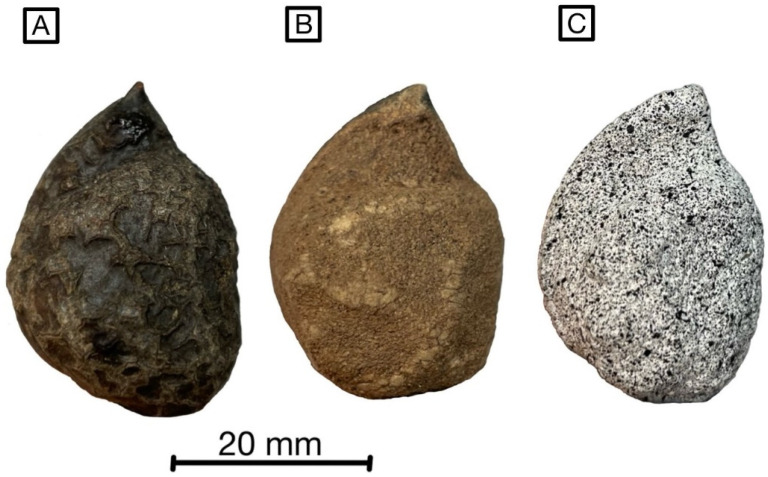
Lateral view of a fruit valve of *Hakea sericea*; with bark and epidermis (**A**), epidermis and bark removed (**B**), with sprayed speckle pattern for DIC analysis (**C**).

**Figure 3 biomimetics-09-00191-f003:**
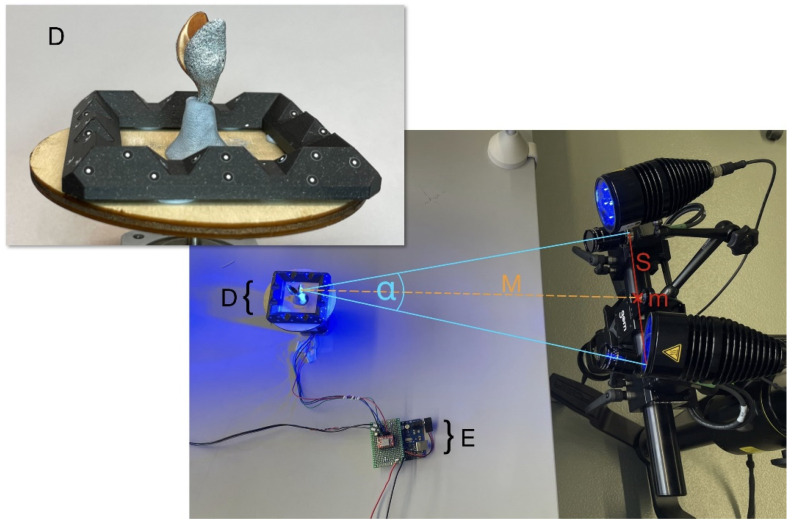
Experimental 360°-DIC setup, consisting of the ARAMIS Adjustable 12M stereo camera system and a rotatable table with sample and reference point frame. The angle α corresponds to the 25° stereo camera angle enclosed by the two camera sensors. The slider distance (S) is the distance between the two camera sensors; the centre point (m) is the centre of this distance. The measuring distance (M) is the distance between the measuring volume and the centre point (m). Further components: rotatable table with sample and reference point frame (D) and control electronics (E).

**Figure 4 biomimetics-09-00191-f004:**
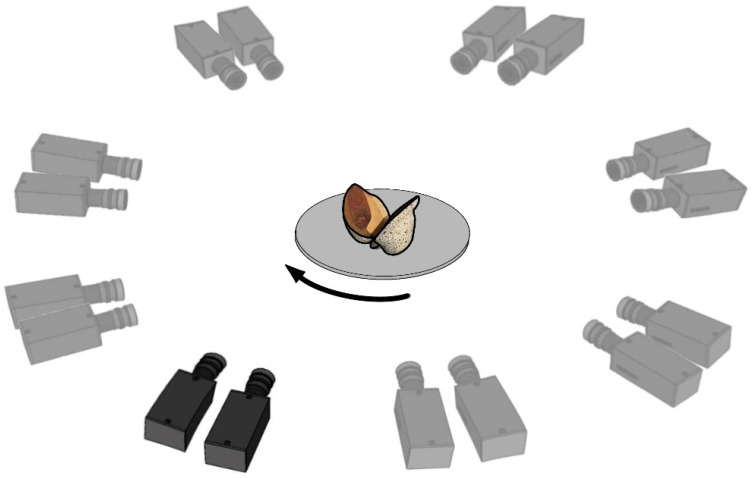
Schematic illustration of the eight perspectives of the 360° DIC, obtained by rotating the *Hakea* fruit in steps of 45 degrees, while keeping the stereo camera system in the same position. This results in one physical position of the cameras (black drawing) and seven virtual positions of the cameras (grey drawing).

**Figure 5 biomimetics-09-00191-f005:**
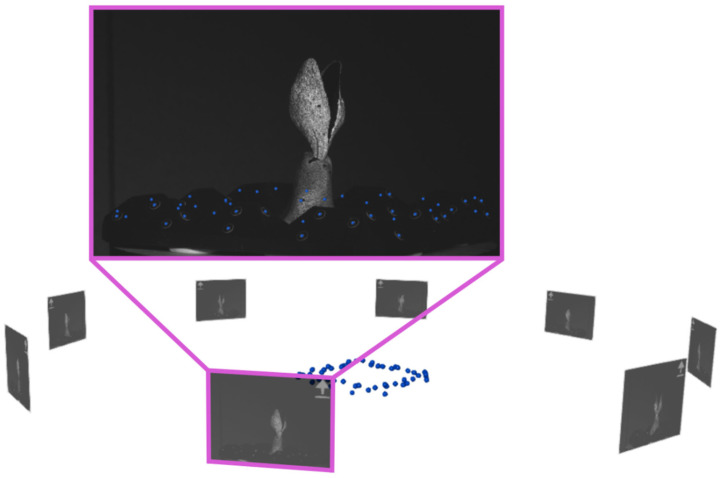
Orientation of eight images using the reference point cloud; the image in the pink frame is shown enlarged as an example. The individual images are orientated in such a way that the visible points in the images correspond almost perfectly to the stored point cloud.

**Figure 6 biomimetics-09-00191-f006:**
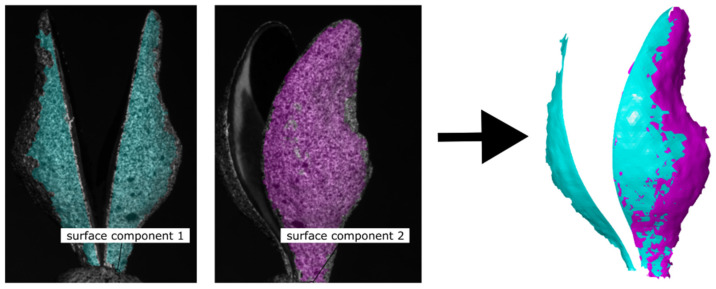
Example of combining two surface components to generate a surface model. Generation of surface components 1 and 2 (**left**), as well as the superimposed surface model (**right**).

**Figure 7 biomimetics-09-00191-f007:**
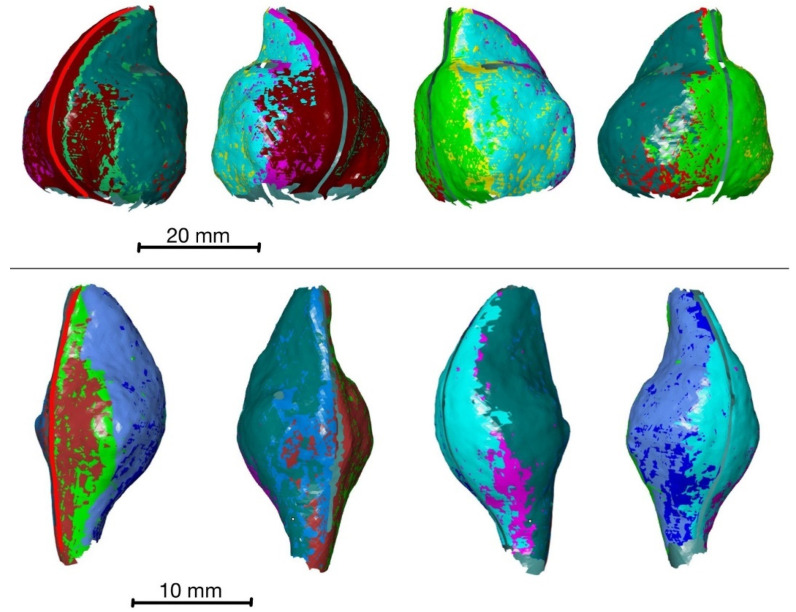
Surface models of *Hakea sericea* (**top**) and *Hakea salicifolia* (**bottom**) consisting of eight superimposed surface components. Different colours are used to indicate the areas of each surface component. Both fruits are shown from the same four perspectives, each rotated by 90°, starting with a 45° perspective on the ventral suture (marked as red line).

**Figure 8 biomimetics-09-00191-f008:**
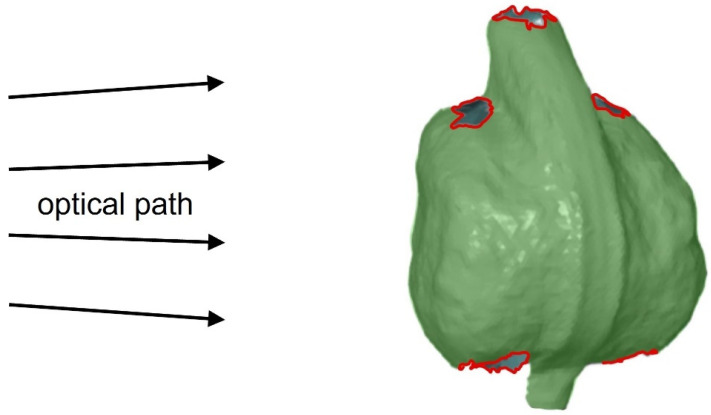
Schematic representation of the lines of view and the resulting undetectable surface areas of a *Hakea sericea* fruit despite using 360° DIC. Areas in red cannot be captured due to the orientation of the camera system. Areas that are clearly visible to both cameras are marked in green.

**Figure 9 biomimetics-09-00191-f009:**
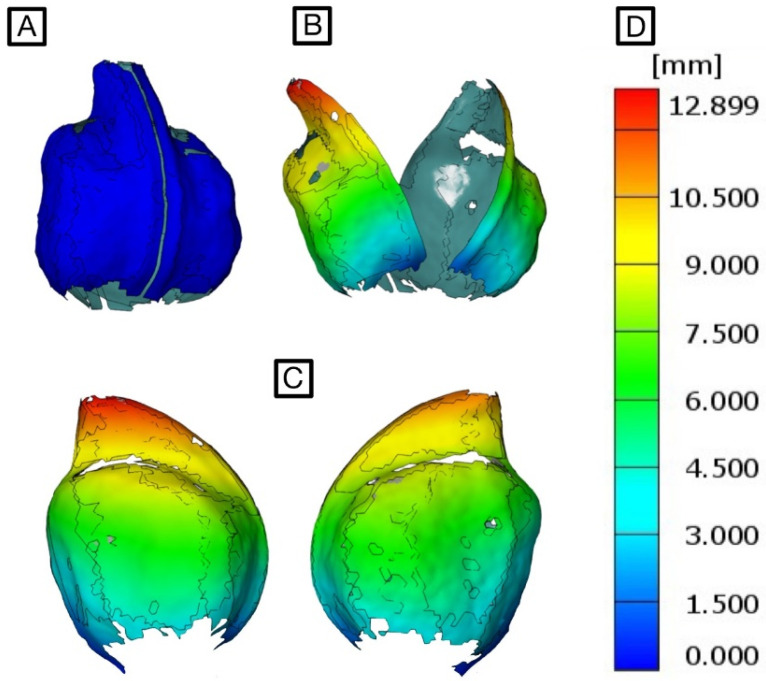
Displacements visualised in colour in the surface model of a *Hakea sericea* fruit: closed (**A**) and open (**B**) state. Detailed view of both valves of the fruit (**C**). The colour-coded displacements correspond to the displacement scale (**D**). Grey lines indicate the edges of surface components.

## Data Availability

The data presented in this study are available on request from the corresponding author.

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
