# Peer review of "Stereo Camera Setup for 360° Digital Image Correlation to Reveal Smart Structures of Hakea Fruits"

_biomimetics, 2024, doi:10.3390/biomimetics9030191_

Round 1
Reviewer 1 Report
Comments and Suggestions for Authors
The paper shows the application of the 360° Digital Image Correlation method, using stereo cameras, to achieve the 3D reconstruction of plants and precise measurement of motion displacement across various regions. The experimental outcomes are highly impressive, but there are still some problems.
1). The Introduction (including the application, principles, limitations of DIC technology, and the implementation of 360°-DIC) contains many redundant descriptions, and it is hoped that further condensation can be achieved to better emphasize the purpose and significance of this research work. For example, lines 76-79, "... binocular images, SEM images, μCT recordings ...", are not necessary and it is suggested to delete them. Moreover, the listed literature requires comprehensive summaries rather than just being enumerated.
2). The authors took great care in acquiring and handling the experimental materials, and provided detailed descriptions. Could the movement of the rotating platform and the pre-experimental treatments (such as spray painting and sticking) potentially affect the opening mechanism of the fruits during the experimental process? Please provide additional clarification and explanation.
3). The 360°-DIC measurement accuracy used is also affected by the rotation accuracy controlled by the stepper motor. What is the motor control accuracy? Is the rotation angle correction based on reference point markers, or can the software automatically identify and correct the angle difference? Please provide additional explanations. Will adding more reference point markers improve accuracy?
4). In line 282, is the ten minutes pause used to process the 360°-DIC results? Why is a full 360° rotation divided into 8 equal 45° steps instead of more positions to improve accuracy?
5). The presented three-dimensional model results are lacking in error evaluation and comparative analysis with alternative 360°-DIC methodologies. Moreover, if the thrust of the article lies in the proposed methodology, a more comprehensive description of the core processing algorithm and its workflow is necessary, rather than a mere mention of "by the GOM Correlate software". Furthermore, given the aim of the study, as stated in the title, "to Reveal Smart Structures", it is imperative to elaborate on the distinctive smart structural characteristics revealed by the experimental findings and highlight the design implications and reference value for smart materials.
Comments on the Quality of English LanguageModerate editing of English language required
Reviewer 2 Report
Comments and Suggestions for Authors
In the paper the authors discuss the solutions for multi-perspective DIC, and present the approach to a 360° DIC system based on a single stereo-camera setup. The desiccation-driven opening mechanism of two woody Hakea fruits over their entire surfaces was characterized. Both the breaking mechanism and the actuation of the two valves in predominantly dead plant material were models for smart materials. An evaluation of the setup for 360° DIC regarding its use in deducing biomimetic principles is given. A way to improve and apply the method for future measurements is discussed.
Please, find below the comments that may improve the quality of the manuscript:
1. Make sure that all acronyms used are explained earlier (e.g. line 77),
2. Captions under the figures are too long, the description should be in the main body of text,
3. Instead 'and colleagues' better use 'et al.',
4. Figure 2 - use (a), (b), and (c) in the description,
5. Line 343 - Figure 9 is twice,
6. Line 297 - Figure 7 is described before Figure 6 - put it in order,
7. Reference 3 - add (in German),
8. Reference 33, 60 - put it in a template style,
9. Conclusions highlighting the novelty of the research are necessary to add.
Comments on the Quality of English Language
Minor editing is necessary (typos); line 109 - rewrite the sentence.
